# Is Permissioned Blockchain the Key to Support the External Audit Shift to Entirely Open Innovation Paradigm?

**Alessio Faccia** [1,*] , **Vishal Pandey** [2] **and Charu Banga** [3]

1    School of Business, University of Birmingham Dubai, Dubai 341799, United Arab Emirates
2    Jindal Global Business School, O.P. Jindal Global University (JGU), Sonipat 131001, India; vpandey@jgu.edu.in
3    School of Business and Law, De Montfort University Dubai, Dubai 294345, United Arab Emirates; charu.banga@dmu.ac.uk
*    Correspondence: a.faccia@bham.ac.uk

**Abstract:** Open Innovation (OI) models have been studied in many fields. However, the challenges and opportunities of a possible OI paradigm application in external auditing have been under-researched. Recent corporate scandals are currently triggering changes and improvements in the regulatory framework by targeting, in particular, the dominance of the so-called "Big Four". The main research question is whether a permissioned blockchain ecosystem could better enhance an OI paradigm and prove more suitable than the Semi-Open Innovation (SOI) paradigm that currently shapes the external audit field. Some challenges are considered in this article. Notably, blockchain requires suitable legal frameworks to ensure legally binding transactions. Moreover, multidisciplinary teams and high investments are required to develop efficient blockchain ecosystems and exploit the power of data analytics. Systematic analysis is performed based on a relevant literature review, along with abductive reasoning and applied modelling methodologies. The analyses demonstrate that the current Semi-Open Innovation external audit model is inefficient because it has led to market concentration, conflicting interests, and even fraud. Therefore, the regulators' role in promoting fully Open Innovation models in the audit industry is essential to ensure transparency, information sharing, fair competition, innovation, and collaboration among audit professionals. Hence, this research aims at providing a different perspective by focusing on the necessary assumptions needed to ensure successful application of technologies in the audit field. The innovative introduction of a permissioned blockchain-based audit system is also suggested to ensure the feasibility of the shift from Semi-Open to Open Innovation.

**Keywords:** open innovation; semi-open innovation; external audit; blockchain; big data analytics; technology; blockchain; audit standards; artificial intelligence; audit regulations; forensic accounting

## 1. Introduction

The adoption of Open Innovation models has been successful in many fields [1–10]. As far as the authors know, no research on OI has yet focused on External Audits.

Innovating the External Audit field can prove challenging because it involves the highest regulated broader accounting domain. The IAASB (International Auditing and Assurance Standards Board) has consistently suggested using technologies support auditors' tasks [11]. The growing attention to innovative technologies is demonstrated by the existence of its subset group, "Technology Workstream Plan" (TWP). This group is currently focusing on the process of identifying, developing and issuing non-authoritative guidance on:

-    "The impact of new technologies on the auditor's documentation;
-    the question about whether an automated audit procedure can be both a risk assessment procedure and a substantive procedure;

- how the nature and number of sources of information affect planning and performing substantive analytical procedures, particularly with the use of data analytic tools." [11]
- The TWG interacts with similar groups set up by other national and international standard-setting boards and committees from an Open Innovation perspective.
- "The International Ethics Standards Board for Accountants (IESBA) established their Technology Working Group in 2018. The IESBA TWG is completing its Phase 1 information gathering and analysis and will present its final report to the IESBA at its December 2019 meeting.
- The Chair of the IAASB's TWG and representatives of Staff have recently engaged with representatives of the PCAOB Office of the Chief Auditor to discuss possible coordination efforts and knowledge sharing about technology in the auditing landscape" [11].

As demonstrated above, forms of collaboration among institutions at the regulatory level are particularly welcome to encourage innovative solutions and the adoption of new technologies. Other forms of collaboration have been observed, and the Big Four (individually, inter-industry) "have already established partnerships, collaborations, or alliances with large technology companies (e.g., Kira Systems, IBM, Accenture)" [12] (See Figure 1). However, there is no evidence of intra-industry collaboration among the audit players (the Big Four in particular) in developing shared innovation strategies and using new technologies. No partnerships, alliances or plans can be found within the same industry to ensure the adoption of the most advanced techniques and technologies in the audit practice.

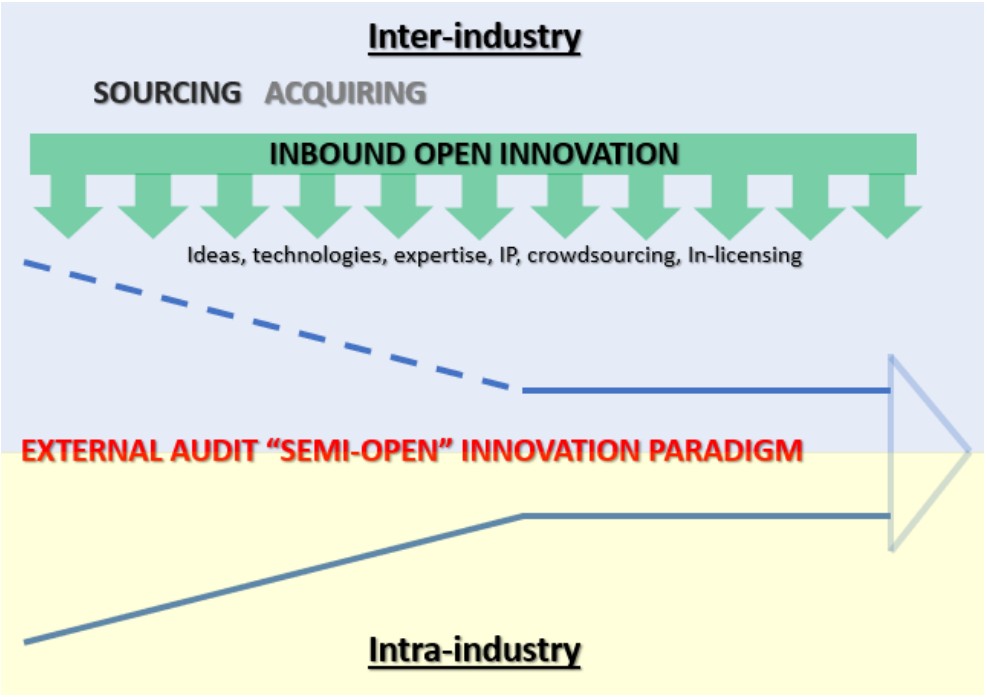

**Figure 1.** External Audit "Semi-Open" innovation paradigm.

This article provides a grounding overview of the innovation landscape that characterises the External Audit field (including advantages and disadvantages of the SOI and OI paradigms applied in the industry) to suggest a feasible model to support the implementation of full OI.

Two consistent research questions and consequent subsets are identified:

RQ1: Is the OI paradigm more suitable than the SOI that currently shapes the External Audit field?

RQ1a: What are the main challenges and opportunities of both OI and SOI paradigms in the context of External Audit?

RQ1b: Given the identified challenges and opportunities, what recommendations can be provided to policymakers, audit firms, and independent auditors?

RQ2: Can a permissioned blockchain platform be suitable for an OI paradigm in External Audit?

Information sharing in criminal and fraud investigations is considered risky and sometimes illegal if confidential information is leaked. Thus, only the sharing of methods, frameworks, and techniques used in audit practice will be considered in this article.

One of the most relevant contributions of this research is the attempt to find a solution to the challenging balance among conflicting interests that arise internally from audit companies and their audited customers (in terms of data confidentiality and privacy) and external stakeholders, investors, and the general public (in terms of trust, lawfulness, and transparency of the financial disclosure) [13]. This is achieved in the proposed model based on a permissioned blockchain that can ensure, at the same time, data confidentiality, privacy, transparency, and compliance with laws and regulations.

Although it is difficult to demonstrate an inverse relationship between audit market concentration and audit quality, as controversial results have been obtained so far [14–17], huge audit failures, inaccuracies, conflicting interests, and scandals have not merely continued to emerge but have increased recently [18–20]. Therefore, the following analyses will be based on abductive reasoning (that allows educated guesswork and conclusions that may be true to be considered best predictions) rather than deductive reasoning (leading to specific conclusions, always true).

Public distrust of the so-called "Big Four" audit companies [21] is leading regulators to reduce their dominance [22]. The limited scalability of audit processes for freelance (and other independent) auditors and small audit companies can be considered one of the main reasons for the current market concentration. In this context, indeed, innovation, especially through new technologies, is often seen by regulators as an opportunity to reduce the gap among the audit players, while it constitutes a trade-off for the Big Four. Undoubtedly, analysing innovation drivers from the two conflicting perspectives (regulators and the Big Four) is another value-added novelty brought by this research.

This paper is structured as follows: (a) identification of materials and methods used; (b) introduction of the theoretical framework, analysis of the relevant literature including R&D investments and companies' size, blockchain and data analytics projects developed by the Big Four, potential impact of blockchain, AI, and big data analytics in the audit field, recent corporate scandals and audit failures; (c) draft of a strategic matrix to perform comparative assessment of challenges and opportunities under SOI and OI paradigms in the external audit market; (d) abductive reasoning conclusions; and (e) identification of suitable recommendations.

The digital revolution continues to massively affect the professional services sector. Data and analytics, artificial intelligence, blockchain, machine learning, and smart automation are innovative technologies that are driving the fourth industrial revolution and the digitisation of business models in the professional services sector. These IT solutions potentially offer almost infinite computational capabilities, automate repetitive tasks, analyse huge data, and develop predictive analyses. These technology solutions represent an extraordinary opportunity for audit companies to improve audit quality significantly.

Value-added applications include unstructured data analyses, such as contracts, e-mails and other documents. Thanks to complex machine learning algorithms, it is possible to develop correlations between very different types of information, which allows proactive highlighting of risks and anomalies, providing insights for improving business processes. From this perspective, the audit truly becomes a strategic activity because these analyses make it possible to exploit all the information potential of the data to support management decisions and improve processes. The fundamentals of professional activity always remain the same: the subject is the professional auditor, who must prepare an opinion on the

financial statements with professional integrity and independence, elements that technology can never replace. Innovation will help obtain further audit evidence to formulate an opinion on the correctness of the financial statements.

For example, among the Big Four audit companies, KPMG has outlined a technological roadmap for the digitisation of auditing activity, which has been developed along four main lines: (a) collaboration: audit teams and their customers communicate and collaborate in real-time to offer the customer a complete view of the progress of the audit activity for a better customer experience; (b) transaction analytics: the ability to examine 100% of transactional information deriving from ERP systems for an overview of company processes and controls; (c) big data analytics: thanks to the computing power of the software and the unlimited availability of information, it is possible to develop predictive analyses; (d) cognitive systems: global alliances with major technology players, such as Google, Microsoft and IBM, combined with knowledge of business models and sectors, allow for the development of continuous learning technologies. Cognitive computing is among the most interesting strands of auditing.

## 2. Materials and Methods

This study is grounded in the following assumptions:

(a)　research and development (R&D) require such a volume of investments that only big corporations can usually afford the cost [23],

(b)　research on blockchain and data analytics applications performed by the Big Four currently follows Open Innovation paradigms, but only when it comes to collaborating with large technology companies [12] and not within the same industry [24,25],

(c)　technology-based innovations focused on blockchain and data analytics can potentially provide great support to audit practices [25–32];

(d)　the alarming recent increase in numbers of sophisticated corporate scandals [33] has demonstrated the inadequacy of current regulations [34,35] and, sometimes, inaccuracies in external audit controls [36–39].

This research is therefore based on:

- Analysis of the most relevant existing literature. It mainly refers to articles published by reliable and reputable sources such as journals listed in Scopus (detailed search criteria and outcomes are presented in Appendix A) and other articles published by the specialised press, such as Bloomberg, Financial Times, and the BBC. A combination of the following keywords was used to search the repositories "Open innovation; External audit; Blockchain; Big data analytics; Technology; Blockchain; Audit standards; Artificial intelligence; Audit regulations; Forensic accounting". More than 200 articles were initially identified. After attempting different combinations, only the most relevant articles (28 in total) were selected and cited in this research, which are included in the references. The selection was made according to the following criteria: (1) Relevance to the studied topic (some articles were not consistent in terms of innovation as they focused on aspects irrelevant to the current aim); (2) When more papers shared the same or similar outcomes, only the most recent was considered; (3) Additional recent information was also considered from other reputable sources such as Fortune, The Times, Financial Times, BBC, EY, PwC, Deloitte, and KPMG.
- The use of an abductive reasoning methodology to draft a subsequent, consistent, and reasoned strategic matrix that compares the Semi-Open Innovation and Open Innovation paradigms. The matrix follows a rigorous approach based on the classification of different criteria and factors that can potentially affect or support innovation in the external audit field. The inference process in abductive reasoning proves particularly suitable while assessing innovating paradigms (inference to best explanation or hypothesis for a set of observations). Abductive reasoning creates tentative explanations to make sense of observations for which there is no appropriate explanation or rule in the existing store of knowledge. It does not start with the explanation but instead links facts together to generate an order that fits the information available—the beginning

of theory building. Successful examples based on this approach are machine learning, design thinking, grounded theory, constructive design research, prototyping, and cultural probes.

- The presentation of a feasible model that includes permissioned blockchain platforms to enhance Open Innovation in the external audit field.

Overall, the research therefore addresses its initial assumptions.

## 3. Results

### 3.1. Analysis of the Relevant Existing Literature

#### 3.1.1. Theoretical Framework

The role of innovation in industrial development has become increasingly central in the public debate, for scholars, regulators, and managers. Economic competition is mainly impacted by innovation, especially for companies operating in challenging and sophisticated environments, which cannot rely on privileged or low-cost access to production factors.

In general, innovators combine different types of knowledge, merging different fields of science and using resources to transform an invention into innovation to exploit it commercially. In the history of economic thought, innovation and technological change have taken on an increasingly strategic role since 1776 when Adam Smith [40] considered the relationship between technological change, division of labour, and structural change in the economy. According to Smith, incorporating technological progress into capital favours the division and specialisation of labour, which is reflected in productivity. Marx emphasised the key role of technology in modern economies and claimed that innovation is a social rather than an individual process [41]. The stimulus for innovation comes from capitalist competitive pressure and the breadth of the markets. Usher [42,43] considered the process of innovation between 1920 and 1929. From Usher's perspective, innovation results from a "cumulative synthesis process" [44], leading from the initial introduction of innovation to its progressive modification and improvement.

However, Schumpeter was the first economist to consider innovation in modern industrial economies broadly, systematically, and in-depth. In his "Theory of Economic Development" [45] in 1911, Schumpeter defined innovation as the main determinant of industrial change, to be considered a creative response of the company to unavoidable change. Innovation can occur in small companies (entrepreneurs' ideas and leadership) and large companies (R&D). Size, however, is not a necessary and sufficient condition for innovation. Schumpeter believes that innovation leads to quick profit, which lasts over time if innovative action is sustained. If this does not happen, the profit disappears due to the reaction of the firms. Innovation, therefore, must be understood as a continuous process of change and the accumulation of knowledge. Economic thought in recent years has focused attention on analysing the characteristics, determinants, and consequences of innovation and technological change. Neoclassical [46] and neo-Schumpeterian/evolutionary [47] models both underline that the scientific and technological opportunities of industries affect the rate of technological progress, and that economic incentives and the appropriateness of results greatly affect companies' innovative efforts. Demand conditions influence the rate of innovation, and there is also a relationship between market structure and innovation. According to the two schools, the degree of concentration of a market structure can generate a certain rate of technological progress.

Among many subsequent contributions in identifying innovation paradigms, the one determined by Henry Chesbrough has assumed particular importance. Chesbrough has theorised the concept of Open Innovation since 2003 in his essay *The Era of Open Innovation* [48]. However, defining so-called Open Innovation is not easy. It is necessary to analyse the phenomenon from both practical and theoretical perspectives. Many companies, especially large ones, have put Open Innovation at the centre of their strategic choices. Adopting this paradigm can bring important advantages, but transitioning from theory to practice is not easy. Nowadays, companies can no longer do without innovation to be competitive. Digitalisation permeates every sector and business activity. Competition is

played out at different levels, and is no longer related only to direct competitors. It is also extended to non-competitors and companies where research is key for progress. Companies in various sectors recognise the value of Open Innovation and use this paradigm with two different approaches: Inbound Open Innovation and Outbound Open Innovation [49].

- Inbound Open Innovation is based on adopting external stimuli to innovate. The most common actions are collaborations with universities and established partners. These involve fewer investments and risks but also more modest results. Other less widespread actions, such as internal incubators and accelerators or the creation of corporate venture capital, have a greater impact on effort and results.
- Outbound Open Innovation involves externalising internal stimuli to undertake innovation actions outside the company. This approach is much less common than the first, which is considered less risky. Furthermore, the most used outbound solutions (joint venture and platform business model) are also the safest in their category because they allow greater intellectual property protection.

Thus, open innovation takes the form of different types of collaboration that can vary in duration and strategic value. Collaborating with one or more start-ups can lead to numerous economic and strategic benefits for all parties, even if it is not always easy to achieve full synergy. Among the most widespread Open Innovation models, the corporate entrepreneurship paradigm assumes that employees are a key asset in terms of accelerating innovation. It aims to enhance entrepreneurial skills and develop new products or services, entering new markets and even independent units.

### 3.1.2. R&D Investments and Company Size

The positive impact of large companies on the wealth of nations has been recognised since Chandler introduced the idea of a "visible hand" [50] to mitigate Adam Smith's blind faith in market equilibrium driven by the so-called "invisible hand" [40,51]. According to many researchers [52–58], the correlation between R&D expenses and business growth (in terms of market capitalisation, revenues, and profits) is evident.

It has been shown that the general conditions for innovation are, moreover, unfavourable to the creation and growth of small businesses (even if they are highly R&D intensive), due to difficulty accessing credit and limited access to the risk capital market [59,60]. The situation is even worse for young entrepreneurs and small innovative businesses [61,62].

Table 1 discloses the seven largest companies in the world by market capitalisation. A high correlation can certainly be noted between the growth rates of R&D expenditures and growth in terms of market capitalisation. Expenses in terms of R&D, considered in absolute value, are enormous. Over the past six years, these seven companies alone have spent more than 533 k million on R&D. In most cases (five out of the seven), the growth rate of market capitalisation is on average higher than the growth rate of R&D expenses.

Investing in research and development requires large investments, and often only large companies can afford it. Given that R&D is essential for growth and that the first movers in a scalable market can benefit from economies of scope and economies of scale, potential new entrants face major barriers to entry [63–65]. The audit market is not immune to this issue [66], where the existence of the Big Four oligopoly ("Big Five" before the demise of Artur Andersen due to the Enron Scandal [67,68]) confirms its concentration.

Extremely simplified, the effects of market concentration are therefore controversial. Although large companies often manage to contribute to the growth of the economy of nations thanks to greater research capacities, from another perspective they create distortions as they reduce the competition. Maintaining a dominant position for too long reduces benefits as large companies may focus on maintaining their dominant position rather than investing in innovation [69–74].

**Table 1.** Biggest companies' capitalisation, R&D expenditures, and growth. Source: Bloomberg.

| Company | Bloomberg | In Millions of USD | FY 2015 | FY 2016 | FY 2017 | FY 2018 | FY 2019 | FY 2020 | Average | R&D |
| Name | Ticker | 12 Months Ending | 30 June 2015 | 30 June 2016 | 30 June 2017 | 30 June 2018 | 30 June 2019 | 30 June 2020 | Growth | Total |
|---|---|---|---|---|---|---|---|---|---|---|
| APPLE | AAPL US | **Market Capitalization** | **639,938.76** | **601,439.27** | **790,050.10** | **1,073,390.54** | **972,268.90** | **1,906,150.95** | | |
| | | *MC Growth* | | **−6%** | **31%** | **36%** | **−9%** | **96%** | 30% | |
| | | R&D Expenditure | 8067.00 | 10,045.00 | 11,581.00 | 14,236.00 | 16,217.00 | 18,752.00 | | 78,898.00 |
| | | *R&D Growth* | | **25%** | **15%** | **23%** | **14%** | **16%** | 18% | |
| AMAZON | AMZN US | **Market Capitalization** | **318,344.19** | **357,687.99** | **566,023.48** | **737,467.27** | **920,224.32** | **1,638,235.79** | | |
| | | *MC Growth* | | **12%** | **58%** | **30%** | **25%** | **78%** | 41% | |
| | | R&D Expenditure | 12,540.00 | 16,085.00 | 22,620.00 | 28,837.00 | 35,931.00 | 42,740.00 | | 158,753.00 |
| | | *R&D Growth* | | **28%** | **41%** | **27%** | **25%** | **19%** | 28% | |
| MICROSOFT | MSFT US | **Market Capitalization** | **354,392.05** | **399,535.36** | **531,312.44** | **757,028.97** | **1,023,856.28** | **1,540,774.21** | | |
| | | *MC Growth* | | **13%** | **33%** | **42%** | **35%** | **50%** | 35% | |
| | | R&D Expenditure | 12,046.00 | 11,988.00 | 13,037.00 | 14,726.00 | 16,876.00 | 19,269.00 | | 87,942.00 |
| | | *R&D Growth* | | **0%** | **9%** | **13%** | **15%** | **14%** | 10% | |
| Alphabet | GOOGL US | **Market Capitalization** | **527,687.37** | **540,170.04** | **729,274.63** | **723,340.70** | **921,949.02** | **1,183,421.09** | | |
| | | *MC Growth* | | **2%** | **35%** | **−1%** | **27%** | **28%** | 18% | |
| | | R&D Expenditure | 12,282.00 | 13,948.00 | 16,625.00 | 21,419.00 | 26,018.00 | 27,573.00 | | 117,865.00 |
| | | *R&D Growth* | | **14%** | **19%** | **29%** | **21%** | **6%** | 18% | |
| FACEBOOK | FB US | **Market Capitalization** | **297,757.70** | **332,724.60** | **512,792.76** | **374,130.86** | **585,373.00** | **778,232.84** | | |
| | | *MC Growth* | | **12%** | **54%** | **−27%** | **56%** | **33%** | 26% | |
| | | R&D Expenditure | 4816.00 | 5919.00 | 7754.00 | 10,273.00 | 13,600.00 | 18,447.00 | | 60,809.00 |
| | | *R&D Growth* | | **23%** | **31%** | **32%** | **32%** | **36%** | 31% | |
| TESLA | TSLA US | **Market Capitalization** | **31,543.31** | **34,523.97** | **52,554.95** | **57,442.28** | **75,717.73** | **677,443.20** | | |
| | | *MC Growth* | | **9%** | **52%** | **9%** | **32%** | **795%** | 179% | |
| | | R&D Expenditure | 717.90 | 834.41 | 1378.07 | 1460.37 | 1343.00 | 1491.00 | | 7224.75 |
| | | *R&D Growth* | | **16%** | **65%** | **6%** | **−8%** | **11%** | 18% | |
| ALIBABA | BABA US | **Market Capitalization** | **193,230.00** | **189,240.00** | **281,370.00** | **446,340.00** | **472,440.00** | **555,045.00** | | |
| | | *MC Growth* | | **−2%** | **49%** | **59%** | **6%** | **17%** | 26% | |
| | | R&D Expenditure | 1598.70 | 2068.20 | 2559.00 | 3413.10 | 5615.25 | 6462.00 | | 21,716.25 |
| | | *R&D Growth* | | **29%** | **24%** | **33%** | **65%** | **15%** | 33% | |
| | | TOT R&D Expenditure | 52,067.60 | 60,887.61 | 75,554.07 | 94,364.47 | 115,600.25 | 134,734.00 | | **533,208.00** |

Innovation is an essential condition for economic and social progress. The current period is characterised by strong evolution and discontinuity from the past due to widespread digitalisation among people and companies.

Therefore, innovation allows companies to improve organisational processes, introduce new products onto the market, and respond adaptively to the constant change in environmental production models.

Countries aim at the innovative dimension as a competitive factor. Every country needs to build an innovation ecosystem that stimulates the efficient use of existing human and financial resources. This requires a new production model more oriented towards innovation that involves all subjects: entrepreneurs, universities, financial institutions, large and small businesses, and public decision-makers.

The use of emerging technologies external to the company is becoming more and more widespread. Companies' financial systems are moving to the Open Innovation paradigm to develop the network to access information, resources, professionalism, skills, and external research results. This enhances their business model—more collaborative environments and interdependencies can be targeted. In perspective, large industries will no longer rely exclusively on research and development activities within company perimeters.

The need to open up to external experiences to accelerate the timing of innovative programs and improve the performance of related investments requires that internal research centres have access to new technologies that quickly ensure added value at limited cost. The goal is to foster the integration of the traditional innovation models of large companies with more advanced solutions. These would be selected through specific challenges between start-ups and innovative high-potential SMEs, using tools such as (a) partnership; (b) shared incubators and accelerators; (c) corporate venture accelerators; (d) spin-in processes; (e) corporate venture capital.

### 3.1.3. Blockchain and Data Analytics Projects Developed by the Big Four

All the "Big Four" audit companies, namely "Deloitte Touche Tohmatsu Limited," "Ernst & Young," "KPMG," and "PricewaterhouseCoopers," are already focusing and investing in the use of both blockchain and data analytics [75–84]. These companies' revenues accounted for about 157 billion USD in 2020 (see Figure 2), far higher than any other industry in the services sector. However, the dominant paradigm in the Audit Market is Semi-Open Innovation, where the players separately incur personnel and investment costs to bring new ideas to the market as quickly as possible to generate profits. Not surprisingly, however, studies carried out by the Big Four on the blockchain (which would bring greater transparency) and data analytics (to improve efficiency) are developing very slowly, therefore showing low interest in innovation, effectiveness, and cooperation [85–96].

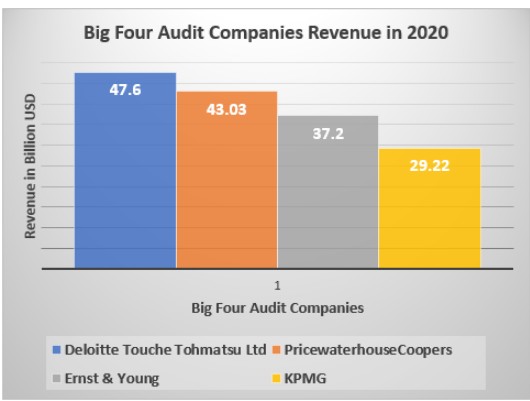

**Figure 2.** Big Four audit companies' revenue in 2020. Sources: Deloitte, PwC, EY, KPMG.

It is now possible to borrow and enhance the perspective provided by the value-added production and distribution model [87] and focus on the publicly available 2020 financial

statements of the Big Four European subsidiaries. Figure 3 astonishingly reveals that no investment was made in R&D by those subsidiaries, therefore suggesting that this function is centralised by the parent company, limiting its diffusion and development.

| | Deloitte (£/000) UK 31.05.2020 | | PWC (£/000) UK 30.06.2020 | | KPMG (£/000) UK 30.06.2020 | | EY (€/000) Netherland 30.09.2020 | | AVERAGE | | |
|---|---|---|---|---|---|---|---|---|---|---|---|
| REVENUE | 2,627,000 | | 4,380,000 | | 2,303,000 | | 859,764 | | | VALUE ADDED CREATED TO BE DISTRIBUTED AMONG CONTRIBUTION FACTORS | P R O D U C T I O N |
| OTHER FIN. INCOME | 80,000 | | 5 000 | | 13,000 | | 1 000 | | | | |
| EXTERNAL EXPENSES | 896,000 | | 1,267,000 | | 628,000 | | 297,286 | | | | |
| VALUE ADDED | 1,811,000 | | 3,118,000 | | 1,688,000 | | 563,478 | | | | |
| R&D | - | 0.00% | - | 0.00% | - | 0.00% | - | 0.00% | 0.00% | INNOVATION CAPITAL | |
| WAGES AND SALARIES | 1,081,000 | 59.69% | 1,951,000 | 62.57% | 1,121,000 | 66.41% | 380,411 | 67.51% | 64.05% | HUMAN CAPITAL | D I S T R I B U T I O N |
| DEPREC./AMORT. | 119,000 | 6.57% | 136,000 | 4.36% | 146,000 | 8.65% | 34,801 | 6.18% | 6.44% | (STRUCTURAL CAPITAL) FIXED ASSETS | |
| EBIT | 611,000 | | 1,031,000 | | 421,000 | | 148,266 | | | | |
| INTEREST EXPENSES | 75,000 | 4.14% | 28,000 | 0.90% | 31,000 | 1.84% | 3958 | 0.70% | 1.89% | (BORROWED CAPITAL) CREDITORS | |
| INCOME TAXES | | 0.00% | 70,000 | 2.25% | 7000 | 0.41% | 751 | 0.13% | 0.70% | (INFRASTRUCTURAL CAPITAL) GOVERNMENT | |
| NET INCOME (LOSS) | 616,000 | 34.01% | 938,000 | 30.08% | 396,000 | 23.46% | 143,558 | 25.48% | 28.26% | (RISK CAPITAL) SHAREHOLDERS | |

**Figure 3.** Big Four audit companies' value-added income statements 2020 (European branches only). Sources: Deloitte LLP UK, PwC LLP UK, EY LLP Netherlands, KPMG LLP UK (The authors have re-classified the companies' disclosed income statements).

In Figure 3, it can also be noticed that most added value is distributed to human capital, represented by the salaries of the companies' employees. This fact confirms that these companies still rely on labour-intensive approaches to perform repetitive tasks, supported by well-developed machine learning and deep learning systems. In the domain of artificial intelligence, both structured and unstructured data can now easily be processed, and this trend could easily be extended to the audit field [88,89]. Although still supervised by human judgment, process automation appears to be more suitable than manual assessment for repetitive tasks based on manuals and detailed rules. Therefore, it is surprising that this industry still displays high wages and salary expenses.

### 3.1.4. Blockchain, AI, and Big Data Analytics' Potential Impact on the Audit Field

According to numerous scholars [90–100], auditing and control activities are among the areas that will be most affected by technologies such as blockchain, artificial intelligence, and data analytics. Indeed, audits involve a highly manual work process, as they process a large amount of unstructured data when identifying risks and controls. As a result, many have suggested the suitability of using artificial intelligence and big data analytics techniques to improve the process [100,101]. Indeed, external audits are exposed to several weaknesses, ranging from the sampling of audit data to inefficient methods and poor staff training. Manual, labour-intensive techniques to identify risks and controls are largely inadequate for assessing and identifying fraud in big corporations.

Currently, the sampling approach generates inevitable approximations. The process then depends on non-measurable human decisions. Risks often remain undetected, and controls are not always adequately associated with the identified risks. The transfer of knowledge is also generally made difficult by the current workload, which leads to a strong dependence on a few key better-trained staff members. Some processes are more exposed to risks than others; however, although the main risk faced in the audit is the financial risk, other critical factors, such as technology, are likely to be overlooked.

Artificial intelligence (AI) and big data analytics can effectively analyse entire datasets, allowing quick outliers and exception identification. As an example, machine learning

could automatically code accounting entries. By creating sophisticated models based on machine learning, auditors could improve fraud detection [94,95].

Blockchain, too, can generate useful changes to audit practices. The duplication of activities and efforts often plagues the current extensive manual work. Usually, the audit process requires auditor to receive and analyse many electronic and manual format documents, and to invest significant time analysing and interpreting the data. This leads to losses in terms of efficiency and cost-effectiveness. While traditional auditing requires ticking and testing transactions and balances on business ledgers at the end of the reporting period, blockchains could ensure the immutable recording of transactions immediately [97,98].

Despite the numerous differences, all blockchains possess the following characteristics:

- they are decentralised peer-to-peer networks in which all network participants keep a copy of the main ledger on their device;
- they keep all the ledger copies constantly updated thanks to the consent protocol.

Blockchain classification aims to identify differences at high levels of application. The distinctions that can be made are related to the advertised dimensions of the network and the presence or absence of access permissions to it. Without too technical a level of detail, the main differences between blockchains are detectable at the level of the participants. It is necessary to identify who is authorised to read all the records on the blockchain authorisation, to sign and maintain the network's cohesion, stability, and integrity (including the mining). There are mainly three types of blockchain: public, permissioned and private. However, this classification is not rigid. Indeed, the characterising elements of these deliniations can be combined to create customised registers for specific applications.

Permissionless or public blockchains require no authorisation to access the network, perform transactions, or to participate in verifying and creating a new block. The most famous are certainly Bitcoin and Ethereum, where there are no restrictions or conditions of access. These are completely decentralised structures, in which anyone can participate, as no central body manages authorisations which are shared among all nodes equally. No network user has privileges over others, no one can control the information stored on it, modify or delete it, and no one can alter the protocol that determines the operation of this technology.

The concepts of permissionless and publicness are closely linked to each other. It would be a contradiction to have a private blockchain where an authorisation is not required to access the recorded data; for this reason, all those that do not require approval are defined as public. Although the data recorded on these blockchains are public, they are encrypted to maintain a sufficient level of privacy. For example, all Bitcoin nodes know the wallet addresses of other users and the transactions that have taken place between them. In principle, these addresses are simply pseudonyms, and unless they are traced to the identity of the real-world person who owns them, a sufficient level of privacy is guaranteed. One method to further protect your identity is to use more than a single wallet address. The main concern related to public blockchains is the issue of scalability, or the ability of a system to improve as the number of participants increases. This type of network is not a scalable technology: as the number of nodes increases, the speed of transactions remains unchanged, but the system's stability increases, thus becoming more secure. Permissioned blockchains are subject to a central authority determining who can access them. In addition to defining who is authorised to be part of the network, this authority defines the roles that a user can cover within it, also defining rules on the visibility of recorded data. Therefore, permissioned blockchains introduce the concepts of governance and centralisation to a network that was originally absolutely decentralised and distributed. A consortium blockchain entrusts the task to a few selected nodes deemed trustworthy, instead of allowing anyone with an internet connection to verify the transaction process.

Specific roles and responsibilities are attributed to accounting and finance; therefore, different authorisation levels should be granted, making the permissionless blockchain not applicable. An external audit can eventually be performed only through a permissioned

platform to ensure confidentiality and relevance of those auditor roles that can oversee and perform the necessary controls.

### 3.1.5. Recent Corporate Scandals and Audit Failures

The UK Financial Reporting Council (FRC) recently found an "unacceptable rate of failures," as well as poor audit quality and practices, in council audits [101]. In the same report, in the "Risk Management" section, it is also stated that the "Audit market is severely disrupted by the failure of a 'Big Four' audit firm or their withdrawal from all or part of the market" [101]. Therefore, growing concerns have led the UK regulator to introduce measures to break up "the dominance of big four firms" [22].

In light of increasingly frequent financial scandals that demonstrate the ineffectiveness of audit controls, the UK regulator has set new rules. By the year 2024, these rules will result in the separation of audit practices from other consulting activities to eliminate bad practices and conflicts of interest that damage audit quality.

None of the Big Four has remained immune from scandals and failures. Table 2 reports, only as examples, some of the most notable cases for these audit companies. The need for change had been evident for a long time. The frequency of audit failure—seemingly without warning signs—has increased in recent years, and Wirecard is the most recent example. Carillion, BHS, Thomas Cook, Patisserie Valerie, and many other companies had each received certificates of good financial health from their respective auditing firms just before collapsing. Therefore, the inadequacy of the auditing sector in carrying out its function was already evident at least a decade ago.

**Table 2.** Two identified recent failures for each of the Big Four audit companies.

| Year of Fraud | Company | Country | Industry | Audit Company |
|---|---|---|---|---|
| 2020 | Wirecard | Germany | Fintech | EY |
| 2020 | NMC Health | UAE/UK | Private hospitals | EY |
| 2018 | Carillon | UK | Construction | KPMG |
| 2010–2013 | Rolls-Royce | UK | Aerospace | KPMG |
| 2014 | Tesco | UK | Retail | PwC |
| 2015–2016 | Redcentric | UK | IT Services | PwC |
| 2010–2011 | Autonomy | UK | IT Services | Deloitte |
| 2018 | Johnston Press | UK | Multimedia | Deloitte |

### 3.2. Comparison of Semi-Open and Open Innovation Paradigms in External Audits

Given the current framework of the audit market and the potential introduction of IT solutions into the audit processes, it was possible to create a matrix of requirements, challenges, and opportunities by comparing Open Innovation and Semi-Open Innovation.

Table 3 summarises the main aspects that can affect blockchain, AI, and big data analytics in the audit field. The main challenges and opportunities are grouped into three perspectives: (a) legal framework, (b) team expertise, and (c) investments. These aspects and related requirements are compared in terms of challenges and opportunities [91,92] faced by the Semi-Open Innovation and Open Innovation paradigms.

The analysis carried out in the previous section proved very useful for preparing Table 3, demonstrating the numerous advantages and comparatively few disadvantages deriving from the introduction of the Open Innovation paradigm in the external audit sector.

**Table 3.** Semi-Open Innovation and Open Innovation paradigms in the external audit market: a strategic matrix.

| Perspectives | Requirements | Semi-Open Innovation ⊗ Challenges ⊘ Opportunities | Open Innovation ⊗ Challenges ⊘ Opportunities |
|---|---|---|---|
| *Legal Framework* | ➤ Ensures privacy (permissioned blockchain)<br>➤ Practices standardisation<br>➤ Enforces fair access to technology | ⊗ Oligopolistic equilibrium<br>⊗ Antitrust concerns due to lack of transparences and standardisation in the audit processes<br>⊗ Difficult standardisation as the progress in innovation is not shared<br>⊗ Progress in innovation slowed by a lack of sharing<br>⊘ Permissioned or permissionless BC are both available | ⊗ Nash equilibrium<br>⊗ Increased complexity of controlling innovation and regulating how contributors affect a project<br>⊘ Increased transparency<br>⊘ Increased standardisation and cooperation among audit companies<br>⊘ Accelerated innovation progress thanks to advancements in sharing<br>⊘ Permissioned or permissionless BC are both available |
| *Team Expertise* | ➤ IT audit Skills<br>➤ Auditors' independence<br>➤ Problem solving skills<br>➤ Strong accounting and finance background | ⊗ IT audit not fully integrated with external audit<br>⊗ Non-collaborative environment<br>⊗ Different goals among departments<br>⊗ High risk of conflicting interests and corruption<br>⊘ Low risk of confidential information leaks | ⊗ Risk of confidential information leaks<br>⊘ Clear and common goals<br>⊘ Collaboration<br>⊘ Transparency and equal opportunities<br>⊘ Low risk of conflicting interests and corruption |
| *Investments* | ➤ IT audit infrastructure<br>➤ Auditors' training<br>➤ Hardware<br>➤ Software | ⊗ High entry barriers<br>⊗ Full affordability is limited to big companies only<br>⊗ Limited integration and diversification<br>⊘ High stimulus to investments in R&D to benefit from intellectual property rights | ⊗ Few stimuli to invest in R&D because results should be shared (no intellectual property rights)<br>⊘ Low entry barriers<br>⊘ Affordability<br>⊘ Integration and diversification |

Open Innovation, in general, is an alternative method of innovation that combines internal and external resources existing in the market, sharing and optimising knowledge. This new approach upsets the classic model according to which innovation must occur within the company to achieve results.

The external audit market is characterised by specific requirements that must be carefully considered. These features are appropriately and consistently reflected in the table, providing a comprehensive and clear understanding of the current specific challenges and opportunities.

## 4. Findings and Recommendations

Table 3 summarises and compares the challenges and opportunities of OI and SOI described in the previous section. Those findings are here supported and explained.

### 4.1. Legal Framework Perspective

In terms of its legal framework, given privacy issues and other regulatory constraints, SOI presents many complications. It ensures an oligopolistic equilibrium, shaped and

demonstrated by the existence of the "Big Four" audit companies. Those companies are very likely to benefit from their dominant position and capital availability to invest in R&D, increasing their advantage [23] in terms of investments. This aspect is crucial for the equitable development of any industry. It has been demonstrated [102–105] that highly concentrated industries cannot ensure sustainable innovation due to leaders' lack of interest, and the lack of affordability for potential small competitors or new entrants.

Creative innovation and intellectual property protection are two essential aspects of business success. There are digital certification proposals thanks to blockchain technology. In theory, creative ideas are the true doorway to the knowledge economy, almost a synonym of innovation. In practice, they remain an asset that is too often poorly protected, sometimes extremely expensive, or inadequate for a complicated world in which digital diffusion knows no boundaries.

There are several tools to protect intellectual property: patents, designs, models, trademarks, and trade secrets, and their appropriate selection usually appears to be the best protection tool to meet the needs of the idea to be protected. Therefore, variables such as the duration of the protection, the type of idea to be protected, and the geographical extent must be considered. More sophisticated solutions are required in external audits, characterised by an oligopoly that determines a trade-off between collaboration and competition.

It must be borne in mind that the right to protection is temporary, because every innovation must become everyone's heritage in the long term. In particular, the patent becomes public, so its use must be carefully evaluated. For example, it will be interesting to evaluate the use of blockchain technology applied to patents to track the filing date with certainty. Intellectual property protection tools can also become business tools, for example, if the owner of the idea does not want to implement it directly but prefers to make it available to others in the form of a license for its use, or to sell the patent itself. Here the dynamics of Open Innovation come into play.

A new strategic and cultural approach should be designed within the OI paradigm. In an oligopolistic environment, information sharing is likely allow companies to maintain their dominant position. It is limited to oligopoly members rather than creating more value and improved competition in the market. The shift expected from all the players in the market (the Big Four included) could be to start transparently sharing ideas, resources, solutions, and tools. In such an environment, all the audit companies could benefit from external technological skills, particularly start-ups, universities, research institutes, suppliers, inventors, programmers, and consultants. Auditors will become aware of the limits of the current proprietary attitude to knowledge, instead considering it a precious asset in an SOI ecosystem. Shared team expertise and national and international regulators could play a leading role in this shift, by creating a shared network and repository that is kept updated with best practices. Indeed, the main target is to regain public trust, affected by many financial scandals that uncovered conflicts of interests and limitations of current audit practices. Public trust is the foundation of financial markets as it is the main factor that pushes people to "convert savings into productive economic growth" [106].

### 4.2. Team Expertise

Statutory auditors must demonstrate various skills:

(a) Financial planning. Fundamental for the auditor is financial planning skill. It requires experience and a solid economic background, and the ability to think about medium and long-term objectives. This aspect involves budgeting and analysis, a process that considers many objective factors [107];

(b) Project management skills. This includes organisation, achievement of objectives, and meeting deadlines. Project work is not easy, especially in the context of auditing. It could be argued that the auditor does not work on projects. However, the application of knowledge, aptitudes, tools and techniques to activities is essential to achieve their objectives. This skill refers to the variables that make up a task: times, costs, and objectives [108].

(c) Analytical reasoning skills. An individual demonstrating this type of reasoning acts critically and knows how to fragment problems to solve them strategically. Auditors with analytical reasoning skills will be able to reduce a problem into steps to be overcome to solve it entirely [109].

(d) Relational and communication skills. This ability is expressed in a particular way within this profession: an auditor's ability to explain himself correctly within the team, and to any clients or superiors.

(e) IT skills. Correct use of IT tools is essential for any professional in the accounting domain; this includes platforms, management systems, the Office package, ERP (and their subsets, AISs—Accounting Information Systems). MS Excel, for example, is a must for those who intend to pursue a career as an auditor: the level of competence required—and essential to carry out the tasks correctly—is very high [110].

(f) Organization and time management. Planning, organisation and—if necessary— reorganisation are essential tasks for those who work with numbers in a team. Furthermore, the auditor must pay great attention to time management and meeting deadlines set by himself and others [111].

Open Innovation may seem more appropriate for large audit companies. Nevertheless, it is an approach that can (and should) be used by all auditors, even small audit companies or freelance auditors. Indeed, teamwork can lead to the identification of clear and common goals, collaboration, transparency and equal opportunities. It also reduces the risk of conflicting interests and corruption.

Researchers are undoubtedly essential for the development of innovation, but innovation must be an integral part of the business culture in order to convert promising ideas into a successful business and ensure that a company becomes and remains truly competitive. It should involve the entire organisation and be the driving force of all its activities. As shown by various international studies, success in innovation is determined more by a company's specific culture and organisational tone at the top than by the size of its R&D investment.

Moreover, implementing the blockchain requires multidisciplinary teams that include transversal skills and knowledge. This scenario will further increase the need for collaborations in different fields to keep up with market requirements [13].

### 4.3. Investments

The lack of successful partnerships between large players and small innovative companies is a structural problem in the audit ecosystem. To facilitate fair competition it is necessary to allow start-ups to grow and to accelerate their access to investments and the audit market. Considerable investments are necessary to meet current requirements for performing an excellent audit, including (a) IT audit infrastructure; (b) auditor training; (c) hardware; (d) software. Although the Open Innovation paradigm might generate reduced stimulus to investment in R&D since the results should be shared (without intellectual property rights), in the case of external auditing, it could bring several benefits in reducing entry barriers and better affordability, integration and diversification.

### 4.4. Theoretical Implications and Recommendations

The theoretical framework introduced in Section 3.1.1 described the impact and importance of technological applications for innovation in any industry. The above results focused on comparative identification of challenges and opportunities for SOI and OI in the external audit field.

The analyses demonstrated that technology drivers in this industry are still underexplored and underused. Despite the industry processes' high suitability for introduction of technology, the current SOI approach might be considered a relevant factor that slows innovation. Therefore, an inbound Open Innovation approach can be suggested to mitigate the risk associated with intellectual property protection. However, after many shameful

scandals, regulators should play a critical role in encouraging the sharing of information and technology to achieve a common goal and restore public trust in the financial markets.

### 4.5. Abductive Reasoning

Abductive reasoning is based on the formation and evaluation of hypotheses using the best available information. In many cases, it is synonymous with "educated hypothesis", an hypothesis process based on a reasoned analysis of available information. Abduction is a reversed process that is used when the rules and the conclusion are known, allowing the best prediction to be identified (although without complete accuracy). This approach considers a specific fact, connects it to a hypothetical rule, and derives an uncertain result from it, a hypothetical conclusion [112].

As already discussed in the introduction, this approach has proved consistent in the analysed case. Evidence, results and best predictions are presented in Figure 4.

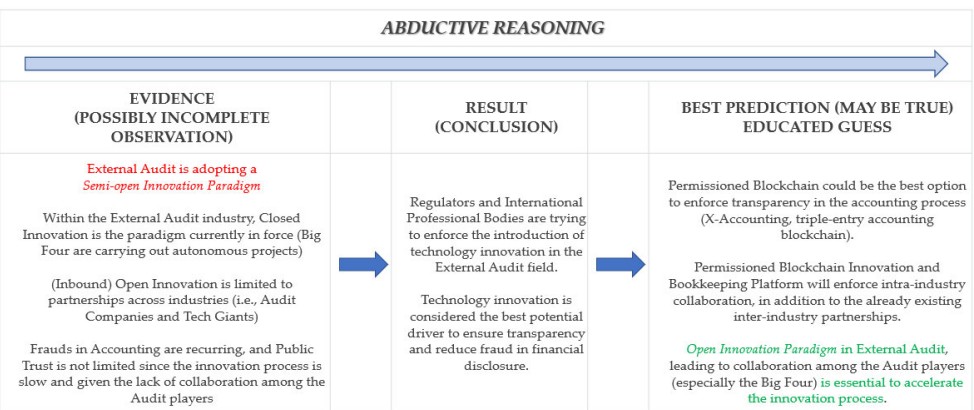

**Figure 4.** Abductive reasoning in external audit innovation paradigms analyses.

## 5. Discussion

This paper addresses the challenges and opportunities of applying an Open Innovation paradigm in the external audit market. Indeed, the need for technological innovation in this industry has been widely considered in the existing literature, suggesting the use of big data analytics [113], blockchain [114–117], and even drones [118]. This industry is currently under scrutiny because of the dominance of the so-called "Big Four" oligopoly that has led to poor auditing quality. As the regulators are now designing and implementing relevant laws and measures to ensure public trust [119–123] by improving quality and transparency, this research can be considered timely in suggesting a new pathway driven by an Open Innovation approach [124].

Compared to the specific requirements of the sector, Open Innovation and Semi-Open Innovation demonstrate complementary advantages and disadvantages, where the opportunities guaranteed by the OI model far exceed those offered by the SOI model. The findings summarised and reported in this research, including the analyses, are supported by extensive literature.

The audit failures of the Big Four, especially when considered in connection with their enormous and growing revenues, generated shameful corporate scandals. This system, therefore, clearly needs a change.

An Open Innovation approach to external auditing, oriented to blockchain, AI, and big data analytics, could offer a valid alternative to the current Semi-Open Innovation model, which has proved unsuitable, highlighting serious transparency, competition, and efficiency problems. This approach should be supported by implementing a permissioned blockchain ecosystem to enhance transparency and provide an audit trail to authorised auditors. This shift, however, is arduous since the participation of regulators, companies, auditors, and other stakeholders in the financial system requires the coordinated implementation of

a blockchain ecosystem. The transparency of a blockchain system combined with the high level of protection and specific authorisation design requested by a permissioned approach can undoubtedly fit the purpose of an Open Innovation paradigm. Indeed, Open Innovation ecosystems could benefit from a reliable, protected system where levels of authorisation secure the information flow, while at the same time the information is stored in a shared ledger.

## 6. Limitations

Although the main benefits of a possible implementation of an Open Innovation paradigm are clear, resistance to its application by the current oligopolists is largely foreseeable, with their ability to exert pressure and carry out lobbying activities. Based on relevant literature analysis, the methodology adopted prepared a matrix comparing OI and SOI applied to the external audit. However, many questions remain open, in particular those related to (a) regulators' desire for paradigm change; (b) the ability of competitors to bridge the gap with the Big Four, (c) commitment of all players to improvement, (d) improvements that can only be appreciated in the long term.

## 7. Conclusions

External auditors ascertain the assets and profits of companies. They are expected to identify and define any corrective errors and frauds (i.e., ensuring that financial statements are fairly disclosed). The professional nature of the external auditor's role has progressively evolved, adapting gradually to new needs, modern organisations and the changing economic climate.

The external auditor now plays an extremely important role in corporate governance by providing constant and careful advice to the company, ensuring independence from the management bodies. A good support for an effective audit starts from rigorous verification and careful accounting, ordered and updated. Therefore, external auditors must always be up-to-date and perform continuous, specialised training.

Modern governance systems involve greater administrative and accounting transparency from companies, favouring reciprocal influence between the various company functions. External auditors can dare to make a concrete contribution to a company regarding its management controls and accounting organisation.

The Big Four oligopoly in the industry could allow audit giants to make necessary R&D investments (given their profitability and expertise); however, no relevant intra-industry collaborations have been observed in their innovation strategies. Partnerships and agreements have been witnessed between the Big Four and the tech giants, shaping a "semi-open inter-industry innovation" approach within the external audit sector. This research compared the current semi-open innovation scenario with a potential fully open-innovation approach from three perspectives: the legal framework, team expertise, and investments. Abductive reasoning helped recognise the potential advantages of complete openness in the industry to accelerate innovation. Moreover, permissioned blockchain was suggested and tested as a practical enabler of this shift by concurrently ensuring transparency and privacy.

**Author Contributions:** Conceptualization, A.F.; methodology, A.F.; validation, C.B. and V.P.; formal analysis, A.F.; investigation, C.B. and V.P.; resources, A.F.; data curation, A.F., C.B. and V.P.; writing— original draft preparation, A.F.; writing—review and editing, A.F., C.B. and V.P.; visualization, A.F.; supervision, A.F.; project administration, A.F.; funding acquisition, V.P., C.B. and A.F. All authors have read and agreed to the published version of the manuscript.

**Funding:** This research received no external funding.

**Institutional Review Board Statement:** Not applicable.

**Informed Consent Statement:** Not applicable.

**Data Availability Statement:** Data available in a publicly accessible repository.

**Conflicts of Interest:** The authors declare no conflict of interest.

# Appendix A

**(TITLE-ABS-KEY "("Open Innovation"") AND TITLE-ABS-KEY "("big data"" OR ""analytics"" OR ""blockchain"" OR ""Artificial Intelligence"") AND TITLE-ABS-KEY (accounting OR audit*))**

De Noronha, T., & Vaz, E. (2020). Theoretical foundations in support of small and medium towns. Sustainability (Switzerland), 12(13) doi:10.3390/su12135312—IRRELEVANT, IT CANNOT BE RELATED TO EXTERNAL AUDIT

Kroon, N., Do Céu Alves, M., & Martins, I. (2021). The impacts of emerging technologies on accountants' role and skills: Connecting to open innovation-a systematic literature review. Journal of Open Innovation: Technology, Market, and Complexity, 7(3) doi:10.3390/joitmc7030163

Lenny Koh, S. C., Genovese, A., Acquaye, A. A., Barratt, P., Rana, N., Kuylenstierna, J., & Gibbs, D. (2013). Decarbonising product supply chains: Design and development of an integrated evidence-based decision support system-the supply chain environmental analysis tool (SCEnAT). International Journal of Production Research, 51(7), 2092-2109. doi:10.1080/00207543.2012.705042—IRRELEVANT, IT CANNOT BE RELATED TO EXTERNAL AUDIT

**(TITLE-ABS-KEY ("Innovation") AND TITLE-ABS-KEY ("audit regulation*" OR "forensic accounting"))**

Edwards, J. R. (1992). Companies, corporations and accounting change, 1835–1933: A comparative study. Accounting and Business Research, 23(89), 59–73. doi:10.1080/00014788.1992.9729861

Hemmelskamp, J. (1997). Environmental policy instruments and their effects on innovation. European Planning Studies, 5(2), 177-194. doi:10.1080/09654319708720392—IRRELEVANT, IT CANNOT BE RELATED TO EXTERNAL AUDIT

House, R., & Musgrave, A. (2013). "Pluralistic 'accountability' for the psychological therapies? holding the tension between 'diversity's virtues and the need for accountability. British Journal of Guidance and Counselling, 41(1), 24-35. doi:10.1080/03069885.2012.750272—IRRELEVANT, IT CANNOT BE RELATED TO EXTERNAL AUDIT

Kralj, D., & Markič, M. (2008). Processes innovation and sustainable development. WSEAS Transactions on Environment and Development, 4(2), 99-108. Retrieved from www.scopus.com—IRRELEVANT, IT CANNOT BE RELATED TO EXTERNAL AUDIT

Rehman, A., & Hashim, F. (2021). Can forensic accounting impact sustainable corporate governance? Corporate Governance (Bingley), 21(1), 212–227. doi:10.1108/CG-06-2020-0269

**(TITLE-ABS-KEY "("external audit"" OR ""external auditing"") AND TITLE-ABS-KEY (innovation) OR TITLE-ABS-KEY (blockchain)) AND (LIMIT-TO (LANGUAGE, "English")) AND (LIMIT-TO (SUBJAREA, "BUSI") OR LIMIT-TO (SUBJAREA, "ECON") OR LIMIT-TO (SUBJAREA, "SOCI") OR LIMIT-TO (SUBJAREA, "DECI"))**

Appelbaum, D., & Nehmer, R. A. (2017). Using drones in internal and external audits: An exploratory framework. Journal of Emerging Technologies in Accounting, 14(1), 99–113. doi:10.2308/jeta-51704

Barr-Pulliam, D., Brown-Liburd, H. L., & Munoko, I. (2022). The effects of person-specific, task, and environmental factors on digital transformation and innovation in auditing: A review of the literature. Journal of International Financial Management and Accounting, doi:10.1111/jifm.12148

Christensen, M., & Skærbæk, P. (2007). Framing and overflowing of public sector accountability innovations: A comparative study of reporting practices. Accounting, Auditing and Accountability Journal, 20(1), 101–132. doi:10.1108/09513570710731227

De Andrés, J., & Lorca, P. (2021). On the impact of smart contracts on auditing. International Journal of Digital Accounting Research, 21, 155–181. doi:10.4192/1577-8517-v21_6

Dyball, M. C., & Seethamraju, R. (2021). The impact of client use of blockchain technology on audit risk and audit approach—An exploratory study. International Journal of Auditing, 25(2), 602–615. doi:10.1111/ijau.12238

Hamdan, S. L., Jaffar, N., & Razak, R. A. (2017). The effect of competency on internal 'auditors' contribution to detect fraud in Malaysia. Paper presented at the Proceedings of the 29th International Business Information Management Association Conference - Education Excellence and Innovation Management through Vision 2020: From Regional Development Sustainability to Global Economic Growth, 1544–1559. Retrieved from www.scopus.com

Hnydiuk, I. V., Datsenko, G. V., Krupelnytska, I. H., Kudyrko, O. M., & Prutska, O. O. (2021). Audit of budget programs in european union countries. Universal Journal of Accounting and Finance, 9(4), 841–851. doi:10.13189/ujaf.2021.090430

Ibáñez, E. M. (2021). Accounting and non-financial firm data tokens in permissioned DLT networks. International Journal of Intellectual Property Management, 11(1), 54–62. doi:10.1504/ijipm.2021.113358

Ji, H. (2020). A periodic auditor designation and the role of audit committee. Global Business and Finance Review, 25(2), 11–18. doi:10.17549/gbfr.2020.25.2.11

Krieger, F., Drews, P., & Velte, P. (2021). Explaining the (non-) adoption of advanced data analytics in auditing: A process theory. International Journal of Accounting Information Systems, 41 doi:10.1016/j.accinf.2021.100511

~~Larner, J., & Mason, C. (2014). Beyond box-ticking: A study of stakeholder involvement in social enterprise governance. Corporate Governance (Bingley), 14(2), 181–196. doi:10.1108/CG-06-2011-0050~~—IRRELEVANT, IT CANNOT BE RELATED TO EXTERNAL AUDIT

Manita, R., Elommal, N., Baudier, P., & Hikkerova, L. (2020). The digital transformation of external audit and its impact on corporate governance. Technological Forecasting and Social Change, 150. doi:10.1016/j.techfore.2019.119751

~~Poddar, P., & Singh, S. K. (2020). INNOVATION and CORRUPTION: DISSECTING CAUSAL LINKAGE USING PATENT APPLICATION INFORMATION from INDIA. Singapore Economic Review, doi:10.1142/S0217590820450046~~—IRRELEVANT, IT CANNOT BE RELATED TO EXTERNAL AUDIT

~~Ronsom, S., & Amaral, D. C. (2017). Evaluation of innovation networks based on sstandardised management system. Gestao e Producao, 24(3), 557–569. doi:10.1590/0104-530X2512-16~~—IRRELEVANT, IT CANNOT BE RELATED TO EXTERNAL AUDIT

Rozario, A. M., & Thomas, C. (2019). Reengineering the audit with blockchain and smart contracts. Journal of Emerging Technologies in Accounting, 16(1), 21–35. doi:10.2308/jeta-52432

~~Ryder, J. (2006). Raising the curtain on culture change. Industrial and Commercial Training, 38(4), 185–189. doi:10.1108/00197850610671955~~—IRRELEVANT, IT CANNOT BE RELATED TO EXTERNAL AUDIT

~~Schmid, J. (2018). Intelligence innovation: Sputnik, the soviet threat, and innovation in the US intelligence community doi:10.1007/978-3-319-75232-7_3 Retrieved from~~ www.scopus.com—IRRELEVANT, IT CANNOT BE RELATED TO EXTERNAL AUDIT

~~Staniškis, J. K., & Katiliute, E. (2016). Complex evaluation of sustainability in engineering education: Case & analysis. Journal of Cleaner Production, 120, 13–20. doi:10.1016/j.jclepro.2015.09.086~~—IRRELEVANT, IT CANNOT BE RELATED TO EXTERNAL AUDIT

~~Work, B. (2002). Patterns of software quality management in TickIT certified firms. European Journal of Information Systems, 11(1), 61–73. doi:10.1057/palgrave.ejis.3000410~~—IRRELEVANT, IT CANNOT BE RELATED TO EXTERNAL AUDIT

~~Wu, J., & Wu, Z. (2014). Integrated risk management and product innovation in China: The moderating role of board of directors. Technovation, 34(8), 466–476. doi:10.1016/j.technovation.2013.11.006~~—IRRELEVANT, IT CANNOT BE RELATED TO EXTERNAL AUDIT

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
