# Peer review of "Is Permissioned Blockchain the Key to Support the External Audit Shift to Entirely Open Innovation Paradigm?"

_2199-8531, doi:10.3390/joitmc8020085_

Round 1

Reviewer 1 Report

Dear Authors, 

I appreciated most of your interventions. They lead to a now complete and significant contribution. My only recommendation is to analyze more the discussion session and to add to this session references in order to enrich it. 

Kind regards

Author Response

Thank you for your valuable feedback. We massively improved the paper and addressed your concerns. 

Thank you once again for your time.

Kind regards,

AF

Reviewer 2 Report

Recommendation: Reconsider after major revision

Comments

I have thoroughly read the entire manuscript titled “External audit, the bold shift from Closed to Open Innovation through permissioned blockchain”, and according to my analysis, the manuscript is characterized by several major issues. In this sense, I have my concerns in regards to its consistency & Systematic Review Study. On which basis the keyword criterion selected (WoS have several filter option) why you have not opted for this. The background is too crowded with no coherence in flow of ideas. My comments are useful for your endeavor.

 Title: your title seems disconnected regarding the blockchain technology to cater audit industry. Perhaps you should link them.

Abstract: there are many things about the abstract which need to be re-evaluated for example the line mentioned “In this research, the authors suggest that, although Blockchain and data analytics are often considered the best way to develop advanced audit practices, they a) need to be implemented in a suitable legal framework, b) by a team of experts with different scientific backgrounds, and c) require a high level of investments.” What does it mean? The open innovation is divided into two sections i.e. closed and open innovation, in this study the existence of the audit industry is not accounted specifically for innovation, therefore, it fails to connect.

Introduction:  I do believe that you need to motivate more your paper, i.e. justify what will be it's value-added. In the introduction, authors mentioned number of paragraphs but this gives no value to the paper, nor helps the reader in understanding the importance and need of this work. Moreover in paragraph 4 mentioned the line “Given that confidentiality and privacy always need to be ensured, public trust, lawfulness, and transparency must also be considered equally important.” what does it mean?. Paragraph 9 mentioned “Their application in the review, for example, makes it possible to overcome the sample analysis of processes and transactions because new technologies can analyze 100% of company data” in which context the argument is done. This highlights that the authors clearly failed to connect their study with existing research. In so doing, the paper doesn't seem to engage with existing knowledge.

Materials and Methods: I have three main concerns here. The first one is related to the steps to follow the systematic literature review. Is the second one related to the keywords you’ve chosen to do your research, i.e. title, and abstract, or elsewhere? Finally, the operator (‘AND’& ‘OR’) used between selected keywords. Systematic literature reviews tend to limit themselves to what is published. As of now, the paper seems to rather complicate the paper.

Results: the goal of a systematic literature review is to map, categorize and classify literature. The objective is to make it easier for further scholars to understand what has been done and what needs to be done. In this sense, I do not think that your paper maps and classifies enough literature. It seems that Figure 1 is in tabular form, but is the author mentioned in figure justify?  In section 3.1.4. What is the basis of figure 4?

 Finding & recommendation: Authors should also make sure that concepts are used in a coherent manner to avoid and reduce the paragraph. Section 4.2 team expertise is not supported by blockchain literature.

Discussion: The discussion and other sections of the paper can only be meaningful after re-arranging the other sections of the paper. A conclusion might elaborate on the importance of the work or suggest applications and extensions Indeed, I invite you to really think it and their utility?

Overall, further attention is needed to revise the manuscript.

All the best!

I hope that my comments will help to improve this manuscript’s impact, quality, and readability.

Author Response

Thank you for your valuable feedback.

It really helped us rethink our research project.  Thanks to your guidance we (believe) massively improved the paper and addressed your concerns.

Attached are the detailed comments and replies.

Thank you once again for your time.

Kind regards,

AF

Title: your title seems disconnected regarding the blockchain technology to cater audit industry. Perhaps you should link them.

Title reshaped into " Is Permissioned Blockchain The Key To Support the External Audit Shift To Fully Open Innovation Paradigm?"

We gave more emphasis to the possible blockchain application considered as the driver of a major shift from semi-open to open innovation in the External audit field.

After further consideration, insights from other experts and additional literature, we identified that the current Innovation paradigm that is shaping the EA industry cannot be considered Closed tout court. It is actually Semi-Open as the audit companies are sharing some information with other entities, but only inter-industry, not intra-industry. This aspect provides an additional layer of novelty to this article, that will become the first to identify semi-open inter and intra industry innovation. (other papers considered inter and intra company semi-open innovation).

Materials and Methods: I have three main concerns here.

- The first one is related to the steps to follow the systematic literature review.

We restructured the article in a way to better balance a mix of methodologies (not only focused on the literature review) since we wanted to overcome the limits of the existing literature.

- Is the second one related to the keywords you’ve chosen to do your research, i.e. title, and abstract, or elsewhere?

We used the following website (Scopus)

https://www.scopus.com/search/form.uri?zone=TopNavBar&origin=sbrowse&display=basic#basic

The search was performed within “Article title, Abstract, Keywords”

- Finally, the operator (‘AND’& ‘OR’) used between selected keywords.

We added Appendix 1 to ensure transparency and completeness of the information provided

- Systematic literature reviews tend to limit themselves to what is published. As of now, the paper seems to rather complicate the paper.

According to our perspective, the combination of an aggregative methodology (such as the literature study) with a transformative method (such as the abductive reasoning) proves extremely efficient in the innovation studies to help demonstrate the blockchain’s role in shifting the external audit from a closed to an open innovation paradigm. The final model of a permissioned blockchain framework is finally tested to confirm the initial assumptions.

The limited quantitative data availability and the complexity of innovation analyses forward-oriented challenge the traditional methodologies based on deductive and inductive reasoning. Therefore, we believe that the study of innovation requires innovative methodologies itself.

Abstract: there are many things about the abstract which need to be re-evaluated for example

- the line mentioned “In this research, the authors suggest that, although Blockchain and data analytics are often considered the best way to develop advanced audit practices, they a) need to be implemented in a suitable legal framework, b) by a team of experts with different scientific backgrounds, and c) require a high level of investments.” What does it mean?

We restructured that part, which evidently was not clear. “Some challenges are considered in this article. Notably, blockchain requires suitable legal frameworks that ensure legally binding transactions. Moreover, multidisciplinary teams and high investments are required to develop efficient Blockchain ecosystems and to exploit the power of data analytics”.

- The open innovation is divided into two sections i.e., closed and open innovation, in this study the existence of the audit industry is not accounted specifically for innovation, therefore, it fails to connect.

The following sentence takes this connection into account “The analyses demonstrate that the current External Audit Closed Innovation model is inefficient since it led to market concentration, conflicting interests, and even fraud. Therefore, the regulators’ role is essential to promote Open Innovation models in the audit industry to ensure transparency, information sharing, fair competition, innovation, and collaboration among audit professionals”, Specific innovations, in particular, those related to the use of blockchain and data analytics in the External Audit field are analysed.

Introduction:  I do believe that you need to motivate more your paper, i.e. justify what will be it’s value-added.

Many parts have been better specified, clarified, or deleted, please see below.

In the introduction, authors mentioned number of paragraphs but this gives no value to the paper, nor helps the reader in understanding the importance and need of this work. Moreover in paragraph 4 mentioned the line “Given that confidentiality and privacy always need to be ensured, public trust, lawfulness, and transparency must also be considered equally important.” what does it mean?

The following part replaced the unclear paragraph:

One of the most relevant contributions of this research is to find a solution to the challenging balance among conflicting interests that arise internally from the audit companies and their audited customers (in terms of data confidentiality and privacy) and external stakeholders, investors and the general public (in terms of trust, lawfulness, and transparency of the financial disclosure). This is achieved by the proposed model based on a permissioned blockchain that can ensure, at the same time, data confidentiality, privacy, transparency, and compliance with laws and regulations”.

Results: the goal of a systematic literature review is to map, categorize and classify literature.

The objective is to make it easier for further scholars to understand what has been done and what needs to be done. In this sense, I do not think that your paper maps and classifies enough literature.

We abandoned the Systematic Literature review that was already similarly performed in

Barr-Pulliam, D., Brown-Liburd, H. L., & Munoko, I. (2022). The effects of person-specific, task, and environmental factors on digital transformation and innovation in auditing: A review of the literature. Journal of International Financial Management and Accounting, doi:10.1111/jifm.12148

The used methodologies are now abductive reasoning, combined with relevant literature. The outcome is the matrix, and the paper is adjusted according to additional findings (semi-open inter/intra-industry innovation)

It seems that Figure 1 is in tabular form, but is the author mentioned in figure justify? 

We initially preferred to include a screenshot (that is why it was considered a figure) of the table to make it more visible. We now turned it back to a table. We cannot do the same with Figure 2 as, despite we tried to do our best the formatting will not be good.

In section 3.1.4. What is the basis of figure 4?

Deleted

Finding & recommendation: Authors should also make sure that concepts are used in a coherent manner to avoid and reduce the paragraph.

We strengthen the article by using more compelling methodologies, better engaging with the existing literature, providing additional contribution to the literature (use of abductive reasoning and introduction of the semi-open inter/intra-industry innovation)

Section 4.2 team expertise is not supported by blockchain literature.

All relevant references were added.

Discussion: The discussion and other sections of the paper can only be meaningful after re-arranging the other sections of the paper.

Done.

A conclusion might elaborate on the importance of the work or suggest applications and extensions

Indeed, I invite you to really think about it and its utility?

Done.

Reviewer 3 Report

The paper focuses on an interesting topic, which falls under the topic of the journal. Unfortunately, the paper presents a series of elements which are not properly stated and which should be better addressed, such as:
  • the assumptions made in the paper are not consistent with the existing scientific literature - please carefully revise this part
  • the contributions of the paper are not well stated and are not presented in comparison with other studies from the field
  • the concluding remarks are too general and do not derive from the results presented in the paper.

Author Response

Thank you very much for your time and valuable feedback. It helped us rethinking about our entire research project and we are glad to resubmit it massively improved. Below are the specific replies to your feedback and attached are the specific answers provided to all the reviewers.

Kind regards,

Alessio Faccia

The assumptions made in the paper are not consistent with the existing scientific literature - please carefully revise this part

The assumption that appeared maybe confusing was the second one, now implemented to better clarify its sense. Indeed the “Big Four” are embracing sometimes Open innovation paradigms (if related to joint research with tech giants), however, they are not cooperating with each other within the same industries, therefore delaying the innovation process.

Another assumption modified is the one related to the Closed Innovation, not reshaped as Semi-Open Innovation.

The contributions of the paper are not well stated and are not presented in comparison with other studies in the field

The most relevant contributions have been now identified in the introduction, where we also specified how this paper stands out from the existing literature.

The concluding remarks are too general and do not derive from the results presented in the paper.

We enhanced massively the concluding remarks by streamlining the specific contributions and improving the general presentation of the findings and the feasible application of the proposed model

My only recommendation is to analyse more the discussion session and to add to this session references to enrich it.

We added more references in the discussion section and enhanced massively the concluding remarks by streamlining the specific contributions and improving the general presentation of the findings and the feasible application of the proposed model

Round 2

Reviewer 2 Report

I have thoroughly read the R1 manuscript and incorporated the changes titled “Is Permissioned Blockchain The Key To Support the External Audit Shift To Fully Open Innovation Paradigm?”. I think some minor changes are required to meet the quality of the journal. In this sense, I have my concerns in regards to its consistency and linked with the External Audit. There is no flow when the author wants cite some paper.

Overall the paper addresses a relevant topic but there are some issues -minor (formatting) - that need to be addressed.

All the best! 

Author Response

thank you once again for your insightful and constructive contribution and for your time.

We had the paper professionally proofread (English spelling and formatting) and we performed some revisions, providing an additional paragraph to better shape the conclusive remarks.

Best regards

Reviewer 3 Report

I encourage you to make minor editions to the style of writing. I recommend you to have a separate paragraph outlining the conclusions of the study, so as to respect the format of a research paper.

Author Response

Thank you very much for your time and always constructive and insightful comments. We had the paper professionally proofread and added the conclusions.

Kind regards